# Determinants of Consumers’ Acceptance and Adoption of Novel Food in View of More Resilient and Sustainable Food Systems in the EU: A Systematic Literature Review

**DOI:** 10.3390/foods13101534

**Published:** 2024-05-15

**Authors:** Monica Laureati, Annalisa De Boni, Anna Saba, Elsa Lamy, Fabio Minervini, Amélia M. Delgado, Fiorella Sinesio

**Affiliations:** 1Department of Food, Environmental and Nutritional Sciences, University of Milan, 20133 Milan, Italy; 2Department of Soil, Plant and Food Sciences (DISSPA), University of Bari Aldo Moro, 70126 Bari, Italy; annalisa.deboni@uniba.it (A.D.B.); fabio.minervini@uniba.it (F.M.); 3Council for Agricultural Research and Economics, Research Centre for Food and Nutrition (CREA-AN), Via Ardeatina, 546, 00178 Rome, Italy; anna.saba@crea.gov.it (A.S.); fiorella.sinesio@crea.gov.it (F.S.); 4Mediterranean Institute for Agriculture Environment and Development & CHANGE—Global Change and Sustainability Institute, University of Evora, 7006-554 Évora, Portugal; ecsl@uevora.pt (E.L.); amdelgado@ualg.pt (A.M.D.)

**Keywords:** alternative protein, edible insects, cultured meat, algae, plant-based food, pulses, mushrooms, neophobia, sensory properties, PRISMA

## Abstract

This review article aims to provide an up-to-date overview of the main determinants of consumers’ acceptance of novel foods (new foods and ingredients) in the EU with emphasis on product’s intrinsic properties (sensory characteristics) and individual factors (socio-demographics, perceptive, psychological) by adopting a systematic approach following the PRISMA methodology. Case studies on terrestrial (i.e., insects, cultured meat and other animal origin products, plant-based food including mushrooms, plant-based analogues, pulses, and cereals) and aquatic systems (i.e., algae and jellyfish) are included focusing on age-related and cross-national differences in consumer acceptance of novel foods and ingredients. General trends have emerged that are common to all the novel foods analysed, regardless of their aquatic or terrestrial origin. Aspects such as food neophobia, unfamiliarity, and poor knowledge of the product are important barriers to the consumption of novel foods, while healthiness and environmental sustainability perception are drivers of acceptance. Sensory properties are challenging for more familiar ingredients such as plant-based food (e.g., novel food made by pulses, mushrooms, cereals and pseudocereals). Results are discussed in terms of feasibility of introducing these products in the EU food systems highlighting strategies that can encourage the use of new ingredients or novel foods.

## 1. Introduction

Sustainable food systems encompass the whole food chain, from the development of sustainable agricultural practices and food distribution systems to the creation of healthy diets and the reduction of food losses and waste. Being one important contributor to global greenhouse gas emissions, food systems play a pivotal role in addressing many, if not all, of the 17 Sustainable Development Goals [1].

Over the past century, enormous progress has been achieved worldwide in improving human well-being, although such progress has come at a considerable cost to the environment. Several global trends are influencing food security and the overall sustainability of food and agricultural systems [2,3].

It is estimated that by 2050, the world and the European Union will face an important food crisis as a direct consequence of population expected growth and climate change. Natural disasters such as floods, storms, extreme temperatures, and droughts, as well as the advance of desertification (e.g., Mediterranean basin) and loss of arable land associated with the increase in sea level and soil salinisation (e.g., North Sea), are only examples of the consequences of climate change, which is already felt [4]. Climate modification, together with changes in agricultural practices and food processing, will induce an expected change in food nutritional quality and in the accessibility of nutritious diets [2,5,6]. Moreover, projected population growth is expected to be concentrated in Africa and South Asia and in the world’s cities. Urbanisation is inevitably accompanied by the transition in dietary patterns and greatly impacts food systems. Higher urban income tends to increase the demand for unhealthy foods, as well as animal-source food [2,7]. Typically, diets are becoming higher in salt, fat, and sugar and are, in general, more energy-dense with consequent increase in overweight and obesity [2,5]. Population ageing is another recognised phenomenon that will inevitably lead to the increase of healthcare burden and economy slowdown with need for personalised nutrition interventions [2,8].

All together, these trends pose a series of challenges to food and agriculture. The international community has recognised these challenges and the need for transformative change to define new sustainable development pathways [2]. Suggested mitigating strategies include investing in alternative protein sources, increasing food shelf-life, fostering food by-products recovery, and enhancing biodiversity by promoting local food, most of which require a substantial change in consumer behaviour [5,9].

This scenario implies several novel foods, both those included in the EU definition of novel food [10] and those commonly consumed and recently provided in food formulations, and preparations with new nutritional functions (e.g., mushrooms as vitamin supplements, plant-based meat substitutes) are entering the market. However, the success of these novel foods still depends to a considerable extent on whether consumers accept those innovations. Consumers are key players in the development of sustainable food systems since their food choices can have an important environmental impact and promote more sustainable food production. Although consumers are increasingly aware of their role in the food system, many people still encounter a variety of barriers to the transition towards sustainable and healthy diets [11]. A thorough analysis of the main drivers and barriers of consumers’ acceptance of novel food is beneficial to assess the feasibility of introducing these products in the EU food system. In this sense, there are numerous reviews dedicated to this topic but most of them deal with a specific novel food, especially insects (e.g., [12,13,14,15]), algae (e.g., [16,17,18,19,20,21,22,23,24,25,26]), cultured meat (e.g., [27,28,29,30,31,32,33,34,35,36,37]), or plant-based analogues (e.g., [38,39,40,41,42,43,44]), while very few provide an overview of different novel foods covering both perceptive and psychological determinants of consumer acceptance [45,46,47,48,49] with only four articles adopting a systematic review approach [45,46,47,50], and none of them including an analysis of the quality of the studies. Another gap in the literature is that existing reviews rarely consider the sensory needs of vulnerable target populations such as children and the elderly [46], and cross-national comparisons are seldom present [45,48].

In view of these limitations, the present review article aims to provide an up-to-date overview of the main determinants of consumers’ acceptance of novel foods intended as new foods and ingredients of animal origin such as insects, cultured meat, and jellyfish, as well as plant-based food including mushrooms, plant-based analogues, algae, pulses, and cereals but also products derived from food by-products or emerging technologies in the EU with emphasis on the products’ intrinsic properties (sensory characteristics) and individual factors (socio-demographics, perceptive, psychological) by adopting a systematic approach following the PRISMA methodology. Additionally, we applied a standardised procedure [51] to perform the quality assessment of selected studies, which was used as a further articles inclusion criterion. Case studies on terrestrial (i.e., insects, cultured meat and other animal origin products, plant-based food including mushrooms, plant-based analogues, pulses, and cereals) and aquatic systems (i.e., algae and jellyfish) are included, focusing on age-related and cross-national differences in consumer acceptance of novel foods and ingredients.

Specific objectives of the present review are as follows: (1) Analysis of the factors that influence acceptance and adoption of novel foods and ingredients in different population groups by geographical area, age class, gender, and socio-economic status in the EU; (2) Assessment of the feasibility of introducing these products in the EU food systems by focusing on consumers’ determinants in buying and eating novel foods; (3) Make a focus on food systems of both terrestrial and aquatic origin to highlight strategies that can encourage the use of new ingredients or novel foods.

This systematic literature review does record data on an aggregate level, and no meta-analysis is performed due to the expected heterogeneity in study design, participant recruitment, outcome, and measurements.

## 2. Materials and Methods

### 2.1. Search Strategy

The literature search was conducted between March and June 2022 consulting Web of Science (Core Collection) and Scopus databases. The following search string was used: (“food choice” OR “food accept*” OR “food liking” OR “food adoption” OR “food attitude” OR “food appreciation” OR “food enjoyment” OR “food rejection” OR “food disliking” OR “food neophobia” OR “food disgust” OR “food purchase” OR “food procurement” OR “food buying”) AND (sustainab* OR environment* OR eco-sustainab* OR ecolog* OR “climate change” OR “greenhouse gas emission” OR “GHG emission” OR GHGE OR footprint OR “novel food” OR “innovative food” OR “alternative food” OR “alternative source”).

The last literature search was done on 15 June 2022 by entering within the “Article title, abstract, keywords” section of both databases the two batteries of keywords. Since one of the overall aims of this review is to provide updated information about new trends in consumer acceptance and adoption of novel foods and ingredients, literature search was limited to studies published starting from 2015 onwards.

### 2.2. Articles Selection

A flow chart summarizing the study selection process is depicted in Figure 1. Three independent researchers conducted the literature search and checked whether there were duplicates. A total of 3462 articles were returned by Scopus (n = 1971) and Web of Science (n = 1491). After excluding duplicates (n = 976), the remaining 2486 articles were screened based on titles and abstracts against inclusion and exclusion criteria (Table 1). Two researchers independently screened the first half of the articles, while the other two researchers screened the second half of the articles. Any disagreement between the researchers was solved by discussion. In case of persisting doubts about eligibility, articles were kept for the following step. Overall, 2312 articles were excluded because they did not meet the inclusion criteria. The number of resulting eligible articles for full-text screening was 174. Two researchers independently screened half the articles, while other two researchers worked on the other half of articles. This phase was associated with data extraction of those articles that were considered eligible for the present systematic review based on inclusion and exclusion criteria and quality assessment (see Section 2.4. for details). Any disagreement between the researchers was solved by discussion. The number of resulting eligible articles used in the present systematic review was 87.

This review is based only on studies conducted in the EU (intended as countries geographically located inside EU borders) as this study is part of the SYSTEMIC project (https://systemic-hub.eu/), “An integrated approach to the challenge of sustainable food systems: adaptative and mitigatory strategies to address climate change and malnutrition”, aimed at implementing adaptive strategies for sustainable food production, consumption, and public health by addressing the diverse impact of climate change on nutrition quality and composition of food and defining standards to achieve food and nutrition security in the EU.

Moreover, as the results of many studies confirm [47,52,53,54,55,56,57,58,59,60], previous consumption habits and familiarity can strongly influence the acceptability of novel foods. Countries that are deeply different in their culture, eating habits, but also social and economic context can hardly be compared, and it should be even more difficult to define common adaptation strategies for sustainable food systems.

**Figure 1 foods-13-01534-f001:**
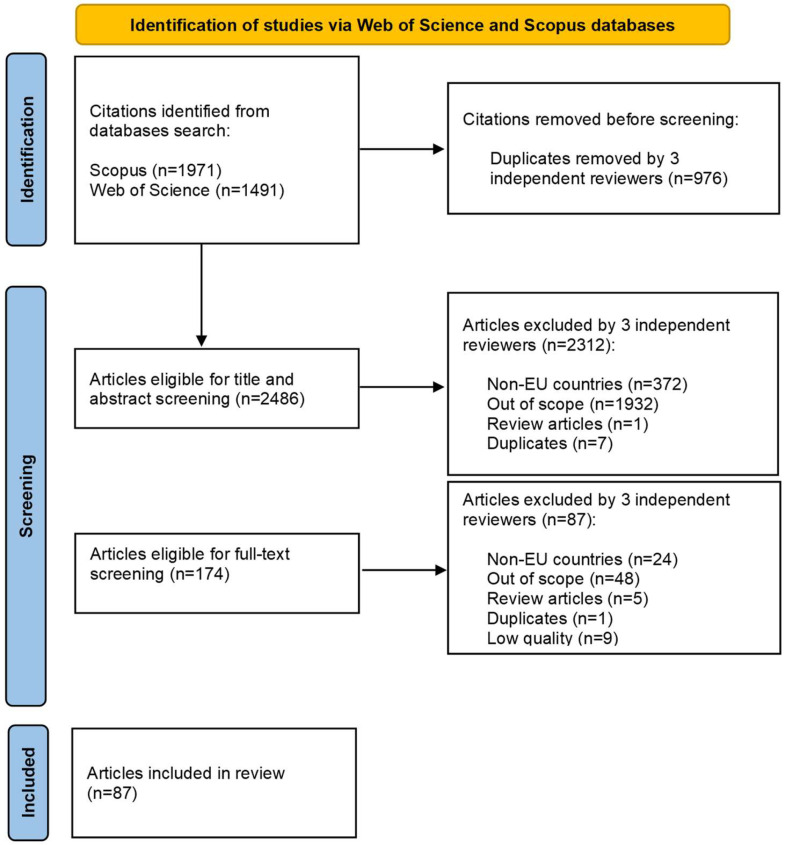
Flow chart depicting the different phases of the systematic review [61].

### 2.3. Data Extraction Process

Data extraction included general citation information (title, authors, year of publication, doi link, database), study characteristics (year of data collection, abstract, objective of the study, sample size, study design/methodological approach), participants’ characteristics (age, gender, country, socio-economic status), determinants of liking/acceptance of novel food explored (psychological traits, individual biological factors, attitudes towards food), outcome (data type, e.g., hedonic, descriptive), findings (type of food/ingredient, main results, conclusion/final remarks). The summary of the main information extracted from eligible articles is shown in Appendix A.

### 2.4. Articles Quality Assessment

A quality assessment of each article (n = 87) was performed following the procedure suggested by Kmet et al. (2004) [51]. The checklist for articles’ quality assessment comprised all 14 criteria: 1. Question/objective sufficiently described? 2. Study design evident and appropriate? 3. Is the method of subject selection described and appropriate? 4. Subject characteristics are sufficiently described? 5. If interventional and random allocation was possible, is it described? 6. If interventional and blinding of investigators was possible, is it described? 7. If interventional and blinding of subjects was possible, is it reported? 8. Outcome measures(s) well defined and robust to measurement/misclassification bias? Means of assessment reported? 9. Sample size appropriate? 10. Analytic methods described/justified and appropriate? 11. Some estimate of variance is reported for main results? 12. Controlled for confounding? 13. Results reported in sufficient detail? 14. Conclusions supported by results?

Each question can be answered with ‘yes’, ‘partial’, ‘no’, and ‘not applicable’. A summary score was calculated for each article considered eligible according to inclusion criteria as follows:
(1)$$\begin{array}{cc} & \mathrm{Summary score}=(\mathrm{Total sum})/(\mathrm{Total possible sum})\\ \mathrm{Where}: & \mathrm{Total sum}=(\mathrm{number of}\;‘\mathrm{yes}’\times 2)+(\mathrm{number of}\;‘\mathrm{partial}’\times 1)\\ & \mathrm{Total possible sum}=28-(\mathrm{number of}\;‘\mathrm{not applicable}’\times 2) \end{array}$$

Although Kmet et al. (2004) [51] did not set a cut-off score below which a given article should be omitted, other authors have done so. According to Henry et al. (2016) [62], who differentiated between strong (>0.8), moderate (0.6–0.8), and weak (<0.6) quality studies, a cut-off score of 0.70 was set. All articles reaching this cut-off score were included in the present systematic review. Each article was evaluated by two reviewers who worked independently and the average score was calculated. In summary, inclusion was firstly based on compliance with the inclusion criteria, and secondly on the achievement of a threshold quality score. Conflicting judgments regarding inclusion of articles were resolved through discussion between the reviewers.

## 3. Results and Discussion

### 3.1. General Description of the Selected Articles

About 30% (n = 26) of the articles reported data from studies conducted in Italy which is the most active country in this research area, while data from the UK and Germany were considered in 16% of the studies and from The Netherlands and Spain in 15% and 11%, respectively.

Concerning the target population, studies included mainly adults (age 18–65 years old, 47.1% n = 41) often extending the age range to older subjects (>65 years, 39% n = 34), while rarely they focused only on specific consumers’ targets such as children and teenagers (age ≤ 19 years old, 8% n = 7) and the elderly (>65 years, 2%, n = 2) or compared children versus adults (1.1%, n = 1). In two cases, participants’ ages were not reported.

Terrestrial products were the most investigated novel foods (76 out of 87 articles, corresponding to 87% of studies), which included mainly insects, other products of animal origin different from insects (mainly cultured meat) and products of vegetable origin (mushrooms, pulses, cereals, upcycled ingredients of vegetable origin) (Figure 2a,b). Aquatic products included mainly algae, while only one study was dedicated to jellyfish.

Psychological traits (food neophobia, food technology neophobia, openness to trying new food, variety-seeking in food choices, anxiety scale, disgust sensitivity, general health interest, food choice motives, environmental concerns, restrained, emotional, and external eating behaviour) and attitudes towards foods (dietary habits, knowledge, perception, and attitudes towards insects, cultured meat, seaweeds, attitudes towards by-products’ reuse, environmental impact of food choices, animal welfare, plant-based protein benefits on health and environment) were explored in, respectively, 63 and 31 articles out of 87, while biological determinants (BMI, taste responsiveness) were rarely addressed (Figure 3). The following sections report the main findings of the selected articles grouped according to the type of novel food investigated: aquatic (mainly algae) and terrestrial (insects, other products of animal origin different from insects, and plant-based products).

### 3.2. Aquatic Systems

Different aquatic products were proposed as potential novel foods in eleven studies. All studies were performed on the adult population, with some of them extending the age range to also include older people (>65 years old) [53,60,63,64,65,66] and children/teenagers [57,65]. No studies were focused only on children/teenagers or elderly people. In total, the percentage of males and females analysed in the different studies corresponds to 51.1% and 48.9%, respectively. Studies were performed in different countries, including Portugal, UK, Spain, Denmark, Germany, The Netherlands, France, Italy and Belgium. Two articles dealt with multi-country comparison [55,64]. All the studies were quantitative, with only one study also presenting qualitative information [66]. Three studies involved tasting of food products [54,55,63]. Food neophobia was the most assessed psychological parameter in the studies.

There were two studies reporting gender differences, with women having higher acceptance of aquatic novel foods, namely seaweeds [52] and spirulina, although in this last case, this age effect was observed only for vegetarians [66]. In the remaining studies, when gender was considered as an explanatory variable, no effects on the acceptance of novel foods from aquatic origin were found. The characteristics of studies on aquatic products are reported in Appendix A.

Like for most novel foods, several studies indicate food neophobia as negatively affecting the acceptance of aquatic novel foods [52,53,55,60,63,64,67]. Nevertheless, some authors also report that this occurs for specific consumer groups, such as foodies, but not for others with different motivations, such as athletes or vegetarians [66]. There was even one study where no effect of food neophobia in the acceptance of sea-buckthorn-based beverage was observed [54].

Taste (and sensory aspects, in general) was observed as being one of the main determinants of acceptance of aquatic novel foods [53,54,55,57,62,64]. In some products, such as those added with spirulina, it was reported that effective masking of its fishy taste increases acceptance [55].

Healthiness perception appears to be a factor influencing the acceptance of these products [52,63,66], as well as familiarity and/or knowledge about the food product [52,53,57,60]. Attachment to meat consumption may also be relevant, with higher beliefs about the benefits of meat and higher meat commitment as barriers to acceptance of aquatic novel foods [53,57,64]. Another interesting aspect, mentioned by some authors, is that more adventurous/open to new experiences/curious consumers are more prone to accept novel foods based on aquatic products [53,57,60]. Although only two studies included consumers from different countries, ref. [55] observed that gastronomic culture was as a factor modulating the acceptance of novel foods of aquatic origin, since there are countries where sensory characteristics associated with these food products are closer to the sensory habits of consumers. For example, a study with spirulina showed France as one of the countries with lower acceptance for incorporation of this aquatic product, suggesting that a reason for this higher resistance is the fact that France has stronger food traditions and culture than countries such as Germany and The Netherlands [55]. In another study, where algae were used to replace meat in burgers, it was also shown that France was characterised by low acceptance for these products [64]. Although French participants perceived these algae-based products as healthier comparatively to meat, this perception was lower than the one expressed by German participants. French participants also perceived these products as having poor sensory quality, resulting in low willingness to replace meat burgers by algae-based burgers.

### 3.3. Terrestrial Systems

#### 3.3.1. Insects

Forty-three studies addressed edible insects as alternative protein foods for human consumption. They were mainly aimed at the adult population (62.8%), although three studies extended the age range to include the elderly (20.9%) and children/teenagers (7.0%). Only three studies were specifically focused on children and teenagers (7.0%), while two articles did not mention participants’ age range. No studies were specifically focused on the elderly population. The characteristics of studies on insects are reported in Appendix A.

Most of the studies (11 articles, 26%) were conducted in Italy, followed by the Netherlands (6 articles, 14%) and Germany (5 articles, 11%). There were three studies conducted in the UK, Poland, and Denmark (7%), and fewer other European countries. The only multinational cross-cultural study surveying nine countries in Europe (France, Spain, The Netherland, UK) and outside Europe (Brazil, China, Dominican Republic, New Zealand, USA) analysed the data as a single global cohort, rather than providing country-by-country analyses; therefore, no conclusion can be driven related to cross-cultural differences for this study [68]. A study was conducted in two countries within Europe (Denmark and Italy) [69], and revealed greater intention to introduce insect proteins into their diet among Danish compared to Italian consumers. Three papers compared a European country with a country outside Europe (Spain and Dominican Republic, Germany and China, and The Netherlands and Thailand, respectively) [70,71,72]. Health and convenience attitudes may drive the adoption of alternative dietary proteins in Spain and the Dominican Republic, with plant-based proteins as preferred alternatives in both countries. Other alternatives in Spain are mycoproteins, cultured meat and insects, while a very negative reaction to the consumption of insects was found among respondents from the Dominican Republic [70]. The study conducted in Germany and China revealed that Chinese participants were more favorable to insect-based food, in terms of taste, nutritional value, familiarity, social acceptance, and willingness to eat than Germans, with no differences between processed and unprocessed food, while Germans reported greater willingness to eat processed insect-based foods compared to unprocessed foods [71]. A qualitative study comparing The Netherlands and a non-European country (Thailand) revealed that Dutch participants, who did not have the same cultural exposure to insects as food as Thais, demonstrated a high level of interest, mainly motivated by the novelty of the experience, environmental and health benefits of eating insects, and their interest in finding sustainable and nutritious alternatives to meat, while Thai participants viewed insects more in terms of taste and familiarity [72].

As regards the methodological approach, almost all studies followed a quantitative approach (98%) mainly managed by online surveys or questionnaires. Focus groups and interviews were the most common methods applied in qualitative studies (20%).

Several psycho-social factors have been explored as variables explaining consumer acceptance of insects as food or predictors of willingness to eat. The factors more studied as determinants of the adoption of insects as food were food neophobia, disgust, previous experience/familiarity, eating behaviour, environmental concern/involvement in sustainability issues, health awareness/concern, and openness to new food experiences. The most frequently studied dependent variables were the intention to try/the willingness to eat/or to buy/the willingness to accept insects as food (59%). Some studies have considered expected liking [73,74,75,76,77] or liking after tasting [78,79,80,81,82,83,84]. Overall, psycho-social factors, such as neophobia and disgust, have been found to be the main barriers to the willingness to try or to accept insect-based products in Europe [68,71,73,76,78,79,81,82,83,85,86,87,88,89,90,91,92,93,94,95,96,97] even though both factors were found to have no significant effects on accepting the idea of insects as meat substitutes [98,99].

On the other hand, previous experience with insects tasting or familiarity with the idea of introducing insects as food were found to increase the willingness to try them [56,71,87,88,94,100,101,102,103] or to positively affect the hedonic evaluation of insect-based products [74,77]. Familiarity was not an important factor in influencing younger consumers’ choice of insect-based food, denoting a major openness [80] and curiosity to novel foods [72,77,104].

Socio-demographic characteristics such as age and gender were found to play a role in the acceptance of edible insects as food in several studies. Age negatively affects the intention to eat insects, with older people being less open [82] or less ready to accept them [76], although children also seem to be not at all inclined to accept products with visible insects [105,106]. Women have greater aversion to eating whole insects than men [76,82,84,92,93] and this discrepancy seems to disappear when invisible insect ingredients are added into processed foods (e.g., pasta, granola, protein bars, jelly sweet). However, it should be emphasised that these outcomes arise from studies using a variety of different experimental procedures and methodologies often reporting conflicting results. Unfortunately, there are few studies that give the opportunity to compare differences by countries and of these even fewer studies that compare results by age and gender [70,71,72]. A study was conducted on university students of two countries within Europe (Denmark and Italy) [69] to investigate the possibility to foster people’s willingness to eat insect-based food through communication, also comparing messages based on individual vs. societal benefits of the eating of insects. The communication effect was significant across nation, gender, and previous knowledge about the topic. Males and people with a higher degree of familiarity were more positive regarding eating insect-based food. Conversely, in another study comparing Spain with the Dominican Republic [70], gender and age were not significant factors for the adoption of insect-based foods as alternative to meat proteins. Finally, no significant effect was observed for gender, age, and education in a cross-cultural comparison between Germany and China [71].

Visibility/appearance of insects was reported being a great barrier to the consumption of insect-based food [72,93]. This has also been demonstrated in studies with children [78]. The authors examined the potential of tactile interactions in the form of a cooking activity to introduce edible insects to children. Two types of insects (grasshopper and mealworm) were incorporated into a snack (oatmeal balls). The grasshopper version of the oatmeal balls received lower hedonic ratings than the mealworm version due to visibility in the oatmeal balls, and higher degrees of animality of these insects.

Protein bars, crisps, and crackers were found to be the preferred carrier (most likely to purchase) for an insect snack product by Western consumers while whole insects are by far the least popular carrier overall, which supports most of the research conducted so far according to which insects are more accepted in the form of a processed ingredient [56]. Indirect entomophagy is also better accepted: La Barbera and coauthors [86] found that the intention to try animals fed with insects and introduce them into the diet was greater than the intention to try dishes based on row insects or processed insects. In general, the visibility/appearance does not seem to be a strongly crucial factor in reducing liking ratings among young people, although in a few studies, some peculiarities might be related to the type of insect. Worms had the lowest score of liking compared to crickets [73]. Spring rolls with visible mealworms were rated as significantly more inappropriate in comparison with spring rolls with invisible mealworms [75].

The hedonic rate increases, and the feeling of disgust decreases, when the product includes ground (crickets and silkworm flour) or processed invisible insects in familiar ready-to-eat preparation (such as in protein bars) rather than whole and visible insects (e.g., chocolate-coated grasshopper and scorpions) [85]. Moreover, the willingness to taste/try them increases when invisible insects are presented as ingredients in familiar formats (e.g., Bolognese, burgers) [90,105] or in protein bars [92]. The invisible inclusion of mealworms had a strong positive effect on acceptability and improved the willingness to try from negative to positive [84]. Biscuits made using insect flour and chocolate-coated grasshoppers were significantly more liked than were other products (cereal bar containing insects, apple salad containing insects, tequila containing a larva, risotto containing maggots, maggot cheese) [76]. Similarly, having a positive tasting experience contributed positively to increasing the degree of the ingredients’ acceptability and food appropriateness [107,108]. Furthermore, exposure to processed insect products (tortilla chips integrated with insect flour) was found to increase consumers’ willingness to consume unprocessed insects [87].

Environmental sustainability followed by health were the most frequent responses regarding reasons for choosing insects as a potential protein source [109], or increasing willingness to try [68,101], even though other studies found that subjects’ involvement in sustainability issues did not play a role in the acceptance of insects [76].

In the Western countries, the consumption of insect is not rooted in the traditional diet, whereas in Eastern countries such as China or Thailand, people are more favourable to all insect-based foods in term of taste, nutritional value, familiarity, and social acceptance [71,72].

Even within Europe itself, there can be different approaches to novel food, due to different food cultures. Although communication can improve attitudes towards insect-based products, the positive effects are less intense in countries where the food cultures and traditions are stronger. For instance, the Italian culture is widely regarded as one of the strongest in Europe. People that have grown up and live in a strong and widely praised food culture may be less susceptible to trying new and different products than people who live in a rapidly changing food culture [69].

#### 3.3.2. Products of Animal Origin Different from Insects

Ten studies addressed the use of non-conventional products of animal origin (different from insects) as novel food. Of these, five studies focused on cultured meat [110,111,112,113,114], while two reported protein- and/or vitamin-enriched meat [115,116], two the use of uncommon foods in modern occidental culture, such as lamb brain and frog meat [107,108], and one a hybrid meat product [117]. Most of the studies were performed with adults (18–65 years), with one study dealing with protein-enriched burgers and involving people more than 65 years old [115], and two studies included children and/or teenagers (9–19 years) [111,112], with these last studies being about cultured meat. Both genders were equally represented in the ten studies. All studies were quantitative, with two of them also presenting a qualitative assessment of sensory quality. Food neophobia was a psychological variable commonly assessed in most studies, except for the study about protein-enriched meat, which assessed food fussiness. Only two studies evaluated the existence of environmental or health concerns as potential determinants of acceptance of novel products of animal origin. The characteristics of studies on other animal products are reported in Appendix A.

Gender (male) and age (older adults) were reported as factors contributing to acceptance of novel alternative foods of animal origin [115]. The role of familiarity in the acceptance of other animal products was not consensual. Although only one study reported familiarity as a significant factor [115], other authors did not observe an influence of familiarity in the consumption of cultured meat [111,114]. Similarly, for disgust, not all studies found the same type of effect, with some observing disgust as contributing negatively to acceptance [110,114] and others not seeing a significant effect [112]. Food neophobia [111,112,113] and food technology neophobia [113,114] were observed as main deterrents of cultured meat acceptance. The different studies also highlighted the relevance of communication in the willingness to consume novel products from animal origin (e.g., cultured meat and unconventional animal products). Claims related to health and sustainability were referred to as effective measures to increase the acceptance of novel foods of animal origin [112,113,115]. For less common ingredients, such as frog meat or lamb brain, the familiarity and labelling indicating food’s edibility and tasting (being aware of sensory characteristics) increased acceptance [107,108]. Moreover, low food neophobia and being male were factors that positively affected the acceptance of this type of products [107].

Some of the studies included in the review were conducted in different countries, reporting differing food acceptance across nations, suggesting that culture may also be a determinant for novel foods of animal origin acceptance [110,113,115]. Comparisons among countries suggest that the ones where people have higher awareness, at different levels (e.g., environmental, health, etc.), and a consequent higher willingness to reduce meat consumption are countries with higher acceptance for meat alternatives [110,115]. It is stressed that countries with strong traditional sensory culture (strong gastronomic influence), and strong food heritage, such as France, are less prone to accept novel foods from animal origin, such as cultured meat, than other countries [115]. People from countries whose food habits have been influenced by different populations (different country and continent origin) may be more open to accept cultured meat [115]. Although the perception of naturalness of cultured meat is, in general, essential for higher acceptance, it may be problematic to generalise the acceptance of this technology from one country to another.

#### 3.3.3. Plant-Based Products

Terrestrial plant-based novel foods have been studied in 22 eligible articles whose main features are summarised in Appendix A.

The target population was represented mainly by adults and individuals aged > 65 years (82% of articles; n = 18), while only one article considered exclusively the elder class of age (>65 years old). Children and teenagers (8–18 years old) were considered in three articles (13.6%).

Most articles (n = 12, 54.5%) focused on Italian consumers, confirming the special attention of Italian scholars on the topic of novel food, generally observed in this review. Data regarding consumers from the UK were reported in 23% of the articles, from Poland in 18% of the articles, and data related to German, Spanish, and Dutch consumers were reported in 14% of articles each. It should be noted that 4 out of 22 articles [115,118,119,120] reported results of studies involving consumers from four or more EU countries. The small number of articles reporting transnational comparison and, above all, the heterogeneity of the products covered do not allow generalisation about the impact of the consumers’ country of origin on the acceptability of novel plant-based foods. However, certain trends can be highlighted. Considering meat consumption and its substitution alternatives [115], consumers from the UK and Finland showed a less favorable attitude regarding meat replacement with plant-based analogues, compared to Polish citizens. Polish consumers were characterised by a positive preference for poultry as the protein source, while they disfavor red meat and plant-based protein burgers. Consumers with the same liking for proteins from red meat, poultry meat, and vegetal origin resulted in almost equal distribution in the UK, Finland, and Poland; meanwhile, a strong negative preference for all three protein sources resulted as more evident for Finnish and Polish consumers. Regarding the attitude towards foods [118] (wheat bread/tomato sauce/consumer potato) obtained from low-impact production processes by microbial applications instead of chemicals, a general high willingness to pay for these products has been observed with no statistically significant effect of respondents’ country of origin. A study considering fungal protein as a meat substitute in Bolognese sauce found that acceptance was related to consumers’ specific characteristics, in particular, food neophobia, attitudes, knowledge, rather than to the country of origin [119]. In a choice experiment study [120], genetically modified apples, cholesterol-lowering peaches and orange juice, and dried black currant with probiotic nutrients have been proposed to consumers from four different countries from Western, Eastern, and Southern Europe. Results showed how social norm information and information about a product’s naturalness may be considered generically effective and less related to the consumers’ nationality having similar effects in all four countries (the Netherlands, Poland, Spain, and Greece). Contrarily, health and product shelf-life information had a different impact on consumers’ attitudes depending on country. Health information was significantly more effective in Greece than in Poland, the Netherlands, and Spain, while for Dutch and Spanish participants, an extended shelf life was preferable compared to Greek and Polish participants.

All studies followed a quantitative approach, mainly managed by online surveys or questionnaires. The most frequent methodological approach was based on the evaluation of sensory properties and consumers’ liking (n = 9, 41%). Willingness to pay (WTP) or accept (WTA) novel plant-based products was assessed in six articles (27.3%), choice experiment methodologies were applied in four works (18.2%) and, finally, three articles (13.6%) evaluated consumers’ perceptions and attitudes towards consumption of novel plant-based food.

The most studied innovative plant-based foods were mainly legume-based foods [121,122,123,124], foods enriched with plant-based by-products [119,125,126,127], and plant-based products specifically developed as meat substitutes [115,117,128,129], followed by novel products made of mushroom and GMO fruits.

Regarding legume-based foods, cookies incorporating fermented grass pea (*Lathyrus sativus*) flour [121], legumes as meat substitutes [122,123], and snack products containing Bambara groundnut flour as an alternative sustainable ingredient [124] were investigated.

Studies concerning foods enriched with by-products investigated acceptability of foods added with upcycled ingredients from olive oil production processes, particularly olive leaves, which are rich in bioactive compounds [130], winemaking by-products [125], bread enriched with upcycled sunflower flour [126], and phenol compounds obtained from unripe grapes for their antioxidant properties [127].

Studies on plant-based meat substitutes focused on consumers’ attitudes in changing their dietary patterns towards lower meat consumption [115], acceptability of meat substitutes (soybeans, mycoprotein, and oatmeal) in traditional meat sauce [128], and on plant-based alternatives to meat in processed meat foods (meatballs, sausages, etc.) [117].

Mushroom-based foods were investigated in three articles [119,131,132] and were tested as a source of proteins [119] and β-glucans [132], as well as innovative and sustainable ingredients for food fortification with vitamin D2 [131].

GMO products were investigated by three articles. More specifically, results about consumers’ attitudes towards fruit innovations improving nutritional values (probiotic contents and cholesterol-lowering effect) were reported by Van’t Riet et al. (2016) [120]. Emotional response towards novel fruits was explored in one article [133] and preference towards low impact cis-genic apples was reported by De Marchi et al. (2019) [134].

Finally, one article [118] evaluated consumers’ WTP for foods (cereals, potatoes, and tomatoes) obtained from innovative agricultural practices using microbial applications to replace synthetic chemical inputs.

Although the appreciation for plant-based foods substantially increased in recent years [135,136], results of this review showed, especially for innovative and unusual products, a great variability in consumers’ acceptance and appreciation. Six articles [115,123,125,126,129,137] stressed the importance of information, underlining the positive effects that awareness of possible health and environmental benefits of plant-based novel foods may have in promoting their consumption. Information was found to be crucial to favour more environmentally sustainable food consumption behaviour [123,129], to encourage the adoption of healthier eating habits [115,137], and to reassure consumers about the healthiness of foods enriched with recycled ingredients and high-value compounds from by-products of the food industry [125,130]. Regarding novel foods enriched with upcycled ingredients from food processes (e.g., pomace and olive leaves, winemaking by-products, upcycled sunflower flour) [125,126,127,129], a better consumer awareness resulted in positively affecting their propensity to consume these products.

Moreover, it has been underlined that information may also contribute to reduce health risks perception and concerns towards new technology and microbial applications used in high-value compounds extracted from by-products and waste and added to food [125,126]. Clear labelling [129,130] and a high level of consumer education [125,130] have led to amplified effects of information, resulting in higher products’ acceptability. Other studies [118,137] also underlined that providing new and complete information may positively affect the WTP for novel foods.

Results from six selected articles [119,121,128,131,132,138] contributed to define sensory features (appearance, texture, taste, etc.) of novel foods. Cereal-based foods (e.g., bread and breadsticks, cookies) with added Pleurotus Ostreatus powder [131,132] and fermented grass pea (*Lathyrus sativus*) flour [121] were acceptable by both children and teenagers. The bread with added Pleurotus Ostreatus powder was even preferred over bread without mushroom powder, especially by teenagers >16 years old [132]. Snacks like whole-grain breadsticks enriched with vitamin D2 from Pleurotus Ostreatus powder [132] resulted as well accepted by children. For sweet and salty cookies [121], ‘tempting’ appearance, ‘crunchy’ texture, ‘sweet’ taste, and ‘odourless’ were features that positively affect children’s preferences, increasing their liking scores. Tartary buckwheat (*Fagopyrum tataricum* Gaertn.) flour supplement added to corn-based porridge-like formulations (called polenta) was in general well accepted but only at low concentrations (up to 30% of addition). The features that were positively associated with polenta samples’ acceptability were related to the low dryness, low intensity of bitter taste, and overall flavour and a low intensity of ochre-yellow colour [138]. In all products (i.e., bread, breadsticks, cookies, and polenta samples), the acceptability progressively decreased with increasing concentration of mushroom powder, fermented grass pea (*Lathyrus sativus*), or Tartary buckwheat (*Fagopyrum tataricum* Gaertn.) flour.

Environmental care (six articles [118,124,126,129,130,137]) has been acknowledged to be a crucial driver for acceptance of novel plant-based foods. Consumers’ awareness and involvement in sustainability issues played a role in choosing novel plant-based foods, especially meat substitutes [115,127,128,129], and contributed to improved acceptability of foods enriched from value-added ingredients from waste and by-products of the food chain [125,126,127,130,139]. Healthcare was reported as a key driver for novel plant-based food acceptability in four articles [118,130,132,137,139]. Consumers declaring a healthy eating behaviour were more prone to appreciate plant-based novel foods with potential health benefits [132].

Psychological traits, in particular, food neophobia and food technology neophobia, were found to be the main barriers to European consumers’ willingness to try or accept novel plant-based foods [121,125,127,128,131,133]. Moreover, unpleasant or unfamiliar sensory attributes of foods and consumers’ ability to sensorially discriminate the new components added to foods [128] were important obstacles to their acceptability, which in some cases [121,127,133] have amplified the rejection effect generated by food neophobia. Notably, the addition of by-products and vegetable waste-based high-value compounds in food formulations was challenging due to their sensory properties that negatively influenced consumers’ acceptance [127]. For instance, the addition of phenols from unripe grapes as antioxidants in beetroot purees [127] affected the perception of sour taste and astringency to a greater extent as concentration increased, resulting in a corresponding decrease in liking.

Overall, consumers’ age had a moderate impact on plant-based novel foods’ acceptability. Nevertheless, elderly people were less prone to accept plant-based foods with high protein content as meat substitutes: namely, European older adults were reluctant to substitute red meat and poultry-based foods with vegetal protein-enriched burgers and declared a negative willingness to pay for protein-enriched burgers if the burgers are plant-based [115]. Consistently, the acceptability of fungal proteins was significantly higher among participants under 35 when compared with those above 35 years [119]. Younger consumers, in particular Millennials [130], were more prepared to accept novel food if their health benefits and environmental sustainability were declared on the label. Finally, older Italian consumers displayed higher liking levels related to corn-based polenta formulations enriched with Tartary buckwheat (*Fagopyrum tataricum* Gaertn.) [138]. This result is probably because polenta samples are considered a traditional food so that incorporation with new plant-based ingredients (i.e., Tartary buckwheat) might be more accepted by older people [140].

In the context of multi-country studies, results from selected articles did not show relevant effects of country distribution of consumers regarding the acceptability of novel foods. Only for protein-enriched burgers [115], Finns were less willing to substitute beef meat with poultry or plant-based protein foods; meanwhile, older Polish adults showed a higher propensity to accept low-impact meats but were indifferent to paying a price premium for protein-enriched burgers.

Finally, the articles considered do not provide a deep exploration of how and to what extent the legal framework, mainly regarding labelling and certification, may improve or modify consumers’ acceptance for vegetable-based novel foods. This should be considered as a research gap to fill by future research.

Considering the high heterogeneity of novel plant-based products, it was deemed appropriate to analyse the strategies to improve acceptability of these products in the EU food systems in reference to the final products that the consumer can find on the market. Table 2 summarises the strategies and points of attention reported in the selected articles which are useful to facilitate the adoption of these novel foods as usual components of diets.

The different studies also highlighted the relevance of communication in the willingness to consume novel products from animal origin (e.g., cultured meat and unconventional animal products). Claims related to health and sustainability are referred to as effective measures to increase the acceptance of novel foods from animal origin, but from the literature review, it is also clear that other specific communication strategies are needed to improve the acceptability and uptake of plant-based novel foods in the dietary habits of European consumers. In general, communication about the health benefits of products enriched with novel ingredients is considered effective in improving the acceptability of fruits and vegetables, especially processed ones [120,137], meat substitutes [115,117,129], bakery products [121,124,126,130,131,139], and food formulations [128,138,139]. Other specific strategies need to be developed in relation to the different final products based on or containing plant-based novel foods. In particular, for fruits and vegetables, research evidence has shown the importance of implementing targeted and effective communication strategies to increase consumer awareness and reduce their skepticism towards new technologies by providing sufficient and understandable information on the use of genetic engineering in food production, as well as in relation to their improved environmental performance [118,120,127,133,134]. In the case of meat substitutes, communication on ethical issues such as animal welfare is also worth implementing [115,117,129]. Last but not least, information that improves consumers’ skills in preparing and consuming novel foods has been reported to improve consumer acceptance [122,123].

Finally, considering plant-based products, general conclusions are difficult to be drawn due to the heterogeneity of studies and products explored. In general, it can be inferred that, as previously reported for the other product categories, food neophobia and information on environmental and health benefits are respectively a barrier and a driver to their consumption. However, in the specific case of new plant-based products, the sensory properties seem to play a predominant role. Aspects such as dark colour, bitter taste, and astringency due to the addition of vegetable ingredients rich, for example, in bioactive compounds such as polyphenols or fibres, are reported to be the main cause of consumer rejection. In this sense, a strategy to mask these aspects can be the addition of flavourings or other sweeteners as well as the blending of different plant-based sources.

As a final remark, it should be emphasised that these general conclusions come from studies using a variety of different tools and methodologies to measure acceptance, intention, or willingness to eat novel foods. It should be considered, indeed, that the results of this review are only marginally comparable because of the plethora of different measurement methods (as well as different population, settings, and different dependent variables) and, in some cases, conflicting results.

## 4. Conclusions

In recent years, a great deal of interest has been devoted to consumer studies in the European context to deepen the knowledge about the availability or intention to consume novel foods and alternative protein sources with attention mainly directed to the analysis of perception towards the consumption of algae, plant-based food, cultured meat, and insect-based foods [47], which are the main focus of the present review. Table 3 summarises the main drivers and barriers of the novel foods and ingredients analysed in the present review.

General trends have emerged that are common to all the novel foods analysed, regardless of their aquatic or terrestrial origin. Aspects such as food neophobia, unfamiliarity, and poor knowledge of the product are important barriers to the consumption of novel foods, while healthiness and environmental sustainability perception are drivers of acceptance.

More specifically, concerning case studies on aquatic systems, algae were the most explored novel food source. Fishy odour and flavour were negative determinants of consumer acceptance. Strategies such as the addition of flavourings, thermal processing, or microencapsulation may contribute to masking these unpleasant sensory properties [19]. A clear pattern of age-, gender- and/or country-related differences cannot be highlighted due to the limited information about micro- and macroalgae as food ingredients.

Concerning insects and insect-based food, the outcome of the present review agrees with recent systematic literature [12,47,141] showing that, in general, in Western cultures, insects are not considered appropriate for consumption and most people are still not ready to add them to daily diets. The present literature review highlights as well that there is a general low willingness of European consumers to eat insects. This likely could be because insect consumption does not belong to the traditional Western diet, which makes the possibility of this novel food becoming part of a habitual Western food pattern complicated in the short term. However, specific consumer targets more willing to adopt insects as food are young males with greater openness to novel food (neophilic subjects) and greater awareness about health (e.g., athletes) and environmental benefits. Moreover, insects may have a higher market success if they are incorporated in invisible form (e.g., as flour) in familiar and traditional food as summarised in Table 3.

Considering products of animal origin different from insects, cultured meat was the most investigated novel food, though only a few studies are available on this topic, and none deal with sensory properties’ perception and actual liking due to legislative obstacles that limit the commercialisation of this novel food in a few countries (United States of America, Israel, and Singapore). Being male and of older age seemed to contribute positively to the acceptance of products of animal origin. No conclusions can be drawn on cross-national differences regarding these products. The different studies also highlighted the relevance of communication in the willingness to consume novel products from animal origin (e.g., cultured meat and unconventional animal products). Claims related to health and sustainability are referred to as effective measures to increase the acceptance of novel foods from animal origin.

Considering plant-based products, general conclusions are difficult to be drawn due to the heterogeneity of studies and products explored. In general, it can be inferred that, as previously reported for the other product categories, food neophobia and information on environmental and health benefits are respectively a barrier and a driver to their consumption. However, in the specific case of new plant-based products, the sensory properties seem to play a predominant role. Aspects such as dark colour, bitter taste, and astringency due to the addition of vegetable ingredients rich, for example, in bioactive compounds such as polyphenols or fibres, are reported to be the main cause of consumer rejection. In this sense, a strategy to mask these aspects can be the addition of flavourings or sweeteners as well as the blending of different plant-based sources.

The present systematic review indicates that communication strategies and awareness campaigns are effective in reducing consumers’ skepticism towards emerging technologies, products, and ingredients by providing sufficient and understandable information related to health and environmental sustainability.

Finally, the large differences observed across different cultures suggest that it may be problematic to generalise findings related to this technology from one country to other countries, and that cross-cultural research may be especially important when it comes to the acceptance of novel food technologies.

The results of this review point to a gap in research regarding transnational as well as cross-age and cross-gender comparisons of factors that favour or hinder the acceptability of novel foods and their wider adoption. Future research should be directed at investigating, for homogeneous product groups, the role that cultural, social, and economic aspects peculiar to each country, beyond individual consumers’ characteristics (neophobia, disgust, liking, etc.), may play in the adoption of novel foods. Moreover, future research perspectives should aim at expanding the geographical scope of the present review including non-EU countries to obtain a global overview of the determinants of novel foods’ consumer acceptance. Future systematic reviews should also be focused on a detailed analysis of the sensory attributes characterizing novel foods to understand their impact on acceptance, possibly including longitudinal studies that can contribute to better understanding changes in consumer acceptance over time. Finally, future research should be dedicated to systematically explore the role that consumer education and awareness campaigns could offer as strategies for promoting novel foods.

Although the major objective of this systematic review was to give a general overview of the main drivers of consumer acceptance of novel foods, it is important to highlight the different implications these new food options may have at different levels, including environmental, social, and economic levels. Until now, most of the research has focused on the environmental impact, supporting that most novel foods/ingredients have a positive contribution to food systems’ environmental sustainability, comparatively to the actual diets highly based on protein from animal sources (e.g., [142]). The social and economic impact of shifting to novel foods is less understood and may be considerably different for different parts of the globe in terms of bringing new economic opportunities, but also new challenges, particularly at the safety and risk assessment level [143].

As a final remark, it should be emphasised that these general conclusions come from studies using a variety of different tools and methodologies to measure acceptance, intention, or willingness to eat novel foods. It should be considered, indeed, that the results of this review are only marginally comparable because of the plethora of different measurement methods (as well as different populations, settings, and different dependent variables) and, in some cases, conflicting results.

## Figures and Tables

**Figure 2 foods-13-01534-f002:**
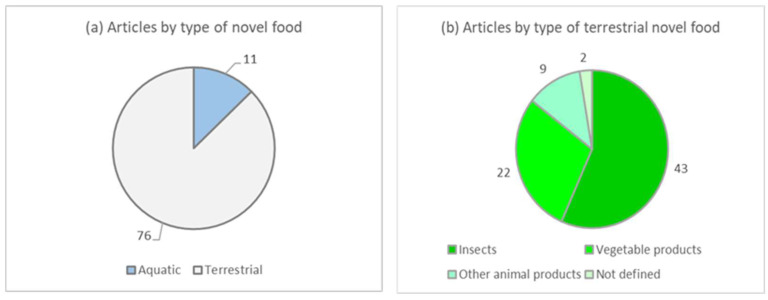
(**a**,**b**) Number of articles by type of novel food (**a**) and by type of terrestrial novel food (**b**).

**Figure 3 foods-13-01534-f003:**
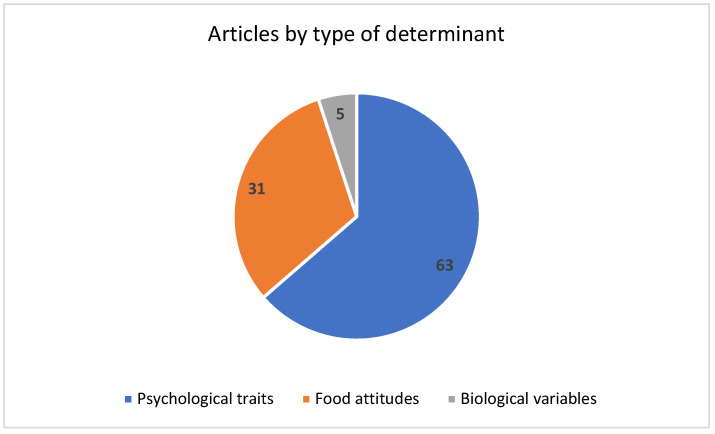
Number of articles by type of determinant explored.

**Table 1 foods-13-01534-t001:** Inclusion and exclusion criteria used for article selection.

Item	Inclusion Criteria	Exclusion Criteria
Participants/population	Studies conducted on individuals of any age and gender	Studies with participants acutely ill or with specific disease; Studies performed in the hospitals or nursing home setting; Studies with participants from non-EU countries (intended as countries geographically located outside EU borders).
Outcome	Both quantitative and qualitative outcomes; Studies analysing the perceptive (e.g., sensory properties) and/or psychological determinants (e.g., food neophobia, disgust) of consumer food choice of novel food.	Studies analysing dietary intake, food consumption frequency, food expenditure; Studies dealing only with marketing or instrumental data.
Study design	No restriction on study design.	Review articles.
Articles’ characteristics	Peer-reviewed journal papers; Studies published in English; Studies published from 2015 onwards.	Grey literature (e.g., thesis, book chapters, reports and conference abstracts).
Articles’ quality assessment	Articles reaching an average quality score ≥ 0.70 (see Section 2.4).	

**Table 2 foods-13-01534-t002:** Strategies facilitating the adoption of plant-based novel foods.

Products	Strategies	References
Novel fruits and vegetables		
Potatoes produced with microbial applications CIS genic apple Genetically modified apples Cholesterol-lowering peaches	Communication of the level of reductions of chemical use and better environmental performances Aspects like well-known products Targeted and effective communication strategies to increase consumers’ awareness and reduce their skepticism about new technologies providing sufficient and understandable information on the application of genetic engineering in food production An adaptation of the regulatory framework to easily distinguish CIS genic products from transgenic is required	[118,120,127,133,134]
Processed fruit		
Dried black currant with probiotic nutrients Cholesterol-lowering orange juice Jam enriched with aloe vera gel	Communication of scientific findings concerning health benefits Information about the positive reactions of other consumers Lowering the selling price of products also through public subsidies to the production of healthy foods	[120,137]
Meat substitutes		
Protein-enriched burgers Plant-based sausages, burgers, vegetable steaks, salami, croquettes, meatballs Veggie-burgers	Communication of scientific findings concerning health benefits Communication of environmental benefit (also through environmental labelling) Communication about ethical issue (animal welfare) Increasing familiarity to novel foods introducing them in public canteens (e.g., kindergarten, school, universities)	[115,117,129]
Products containing soybeans (e.g., soya milk, grains, tofu cottage cheese, half-products, e.g., soya chops)	Communication of scientific findings concerning health benefits Information improving consumers’ knowledge and skills in preparing and consuming pulse-based foods	[122,123]
Bakery products		
Biscuits made with upcycled ingredients Breadstick enriched with mushroom powder Salty and sweet cookies enriched with fermented grass pea flour Salted taralli, crackers, and breadsticks enriched with olive leaves’ extracts. Biscotti and two crackers made from Bambara	Information about product origin Communication of health benefits of product enriched with novel ingredients Communication of environmental benefit of product enriched with upcycled ingredients (also through environmental labelling) Increasing familiarity to novel foods introducing them in public canteens (e.g., kindergarten, school, universities) Carefully proportionate the amount of novel ingredient in order not to excessively alter the appearance and taste of the original product	[121,124,126,130,131,139]
Food formulation		
Bolognese Polenta from pseudocereal Tartary Buckwheat (gluten-free) Mayonnaise enriched by olive leaves’ extracts	Appearance and taste as similar as possible to well-known traditional preparation Communication of health benefits of product enriched with novel ingredients Adding innovative components to formulations that consumers are used to	[128,138,139]

**Table 3 foods-13-01534-t003:** Summary of the main drivers and barriers of the novel food and ingredients analysed in the present review.

Type of Novel Food	Drivers	Barriers
Aquatic products (algae)	Familiarity with the product	Food neophobia Unpleasant sensory properties (fishy odour)
Insects	Familiarity/Previous experience Age (being young) Gender (being male) Information about environmental and health benefits Technological aspects (insects as flour or in invisible form) Type of food product (familiar and traditional food)	Food neophobia and disgust Technological aspects (whole insects in visible form)
Other animal origin products	Gender (being male) Age (being older) Claims about health and environmental sustainability	Food neophobia and food technology neophobia
Plant-based products	Information about environmental and health benefits	Food neophobia Sensory properties (appearance, bitterness, astringency, texture)

## Data Availability

No new data were created or analyzed in this study. Data sharing is not applicable to this article.

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
