# Peer review of "Determinants of Consumers’ Acceptance and Adoption of Novel Food in View of More Resilient and Sustainable Food Systems in the EU: A Systematic Literature Review"

_foods, 2024, doi:10.3390/foods13101534_

Round 1
Reviewer 1 Report
Comments and Suggestions for Authors
1.Please explain or describe the [the achievement of a threshold quality score] in the section of article quality assessment.
2.Is there any possibility to compare the results from Aquatic systems vs Terrestrial systems? If it is possible, will suggest so.
Author Response
We thank the reviewer for the valuable comments. Our reply are reported in attachement in red font. Please see the attachment.

Reviewer 2 Report
Comments and Suggestions for Authors
The manuscript titled "Determinants of consumer’s acceptance and adoption of novel food in view of more resilient and sustainable food systems in the EU: a systematic literature review" provides a comprehensive overview of factors influencing consumer acceptance of novel foods, including product intrinsic properties and individual factors. This review is timely and relevant, given the growing interest in sustainable food sources. However, it could benefit from further clarification to enhance the manuscript's contribution to the field.
1. The manuscript covers a range of novel foods but does not always clearly define or distinguish between them. For example, distinctions between alternative protein food products like insects, cultured meat, and plant-based foods could be made more explicit. Clarifying these definitions early in the manuscript will help readers understand the scope and focus more directly and clearly.
2.The authors should tell us more about the rigor and score calculation results of the selected literature to help readers understand the methodology.
3.While the manuscript touches on age-related and cross-national differences in consumer acceptance, these aspects could be explored in more depth. Specifically, how cultural backgrounds and demographic factors influence perceptions and acceptance of novel foods warrants further discussion. Incorporating more detailed analysis or case studies from different EU countries could provide richer insights.
4.The role of consumer education and effective communication strategies in enhancing acceptance of novel foods is crucial but not adequately covered. The manuscript could benefit from a section that reviews successful communication strategies and educational interventions, providing guidance for stakeholders in promoting novel foods.
5. The manuscript mentions environmental sustainability as a driver of novel food acceptance but does not delve deeply into the sustainability implications of various novel foods. A more thorough examination of the environmental, ethical, and social implications of producing and consuming these foods could add valuable context to the discussion on consumer acceptance.
6.The manuscript concludes with general observations but could offer a more robust discussion on gaps in the current literature and specific recommendations for future research. Identifying under-researched areas, emerging novel foods, and methodological approaches would guide future studies in this evolving field.
Author Response

(The authors gave the same response as above.)

Reviewer 3 Report
Comments and Suggestions for Authors
This provides a comprehensive overview of the determinants of consumer acceptance and adoption of novel foods in the European Union (EU), with a focus on promoting more resilient and sustainable food systems. The authors utilized a systematic approach following the PRISMA methodology to analyze a wide range of studies that investigate consumer preferences for novel foods, including terrestrial and aquatic products, such as insects, cultured meat, plant-based foods, algae, and jellyfish.
Key findings from the review suggest that general trends across all novel foods, regardless of their origin, include food neophobia, unfamiliarity, and poor knowledge of the product as significant barriers to consumption. On the other hand, perceptions of healthiness and environmental sustainability are major drivers of acceptance. The review highlights that sensory properties are particularly challenging for more familiar ingredients like plant-based foods, where novel foods made from pulses, mushrooms, cereals, and pseudocereals are explored.
The review also addresses age-related and cross-national differences in consumer acceptance, indicating that these factors play a role in the willingness to try and adopt novel foods. However, the review notes that further research is needed to understand these differences better and to develop strategies that can encourage the use of new ingredients or novel foods in the EU food systems.
The paper contributes valuable insights into the factors influencing consumer acceptance of novel foods in the EU and provides a foundation for future research and policy-making aimed at promoting sustainable food systems.
However, it could be improved by expanding its geographic scope to include non-EU countries for a global perspective, including more recent studies for updated insights, and providing a detailed analysis of sensory attributes to understand their impact on acceptance. Incorporating longitudinal studies could help understand changes in consumer acceptance over time, while exploring the role of consumer education and awareness campaigns could offer strategies for promoting novel foods. Analyzing economic factors such as price and affordability, including studies that assess consumer perception of sustainability, ensuring diversity among participants, conducting cross-cultural comparisons, and integrating qualitative insights could all contribute to a more comprehensive understanding of consumer acceptance of novel foods.
Comments on the Quality of English Language
No comments.
Author Response

(The authors gave the same response as above.)
